# Mapping Two Decades of New York State Forest Aboveground Biomass Change Using Remote Sensing

Haifa Tamiminia [1,*], Bahram Salehi [1], Masoud Mahdianpari [2], Colin M. Beier [3] and Lucas Johnson [4]

1 Department of Environmental Resources Engineering, State University of New York College of Environmental Science and Forestry (ESF), Syracuse, NY 13210, USA
2 CORE and Department of Electrical and Computer Engineering, Memorial University of Newfoundland, St. John's, NL A1B 3X5, Canada
3 Sustainable Resources Management, State University of New York College of Environmental Science and Forestry (SUNY-ESF), Syracuse, NY 13210, USA
4 Graduate Program in Environmental Science, State University of New York College of Environmental Science and Forestry (SUNY-ESF), Syracuse, NY 13210, USA
* Correspondence: htamimin@esf.edu

**Abstract:** Forest aboveground biomass (AGB) provides valuable information about the carbon cycle, carbon sink monitoring, and understanding of climate change factors. Remote sensing data coupled with machine learning models have been increasingly used for forest AGB estimation over local and regional extents. Landsat series provide a 50-year data archive, which is a valuable source for historical mapping over large areas. As such, this paper proposed a machine learning-based workflow for historical AGB estimation and its change analysis from 2001 to 2019 for the New York State's forests using Landsat historical imagery, airborne LiDAR, and forest plot data. As the object-based image analysis (OBIA) is able to incorporate spectral, contextual, and textural features into the regression model, the proposed method utilizes an OBIA approach and a random forest (RF) regression model implemented on the Google Earth Engine (GEE) cloud computing platform. Results demonstrated that there is a considerable decrease of $983.79 \times 10^6$ Mg/ha in the AGB of deciduous forests from 2001 to 2006, followed by an increase of $618.28 \times 10^6$ Mg/ha from 2006 to 2011, continued with an increase of $229.12 \times 10^6$ Mg/ha of deciduous forests from 2011–2016. Finally, the results demonstrated a slight change in AGB from 2016 to 2019. The transferability of the proposed framework provides a practical solution for monitoring forests in other states or even on a national scale.

**Keywords:** object-based image analysis; Landsat imagery; airborne LiDAR; change detection; state-wide mapping

## 1. Introduction

Forest covers about 31 percent of the world's total land area, the largest natural land-based ecosystem on Earth [1]. Since 1990, the world has lost an estimated 178 million hectares of forest [1]. The rate of net forest loss decreased from 1990 to 2020 due to the reduction of deforestation in some countries and the increase in forest area in others on account of afforestation and the natural expansion of forests [1]. In spite of this, forest loss continues to be of concern, particularly given increasing rates of carbon emissions across the globe and the ability of forest ecosystems to absorb carbon emissions. Forests absorb carbon dioxide ($CO_2$) from the atmosphere through the photosynthesis process. This absorbed carbon is integrated into various biophysical aspects of trees in the form of biomass, both dead and alive, including branches, trunks, and leaves [2]. Thus, providing accurate aboveground biomass (AGB) estimations is of paramount importance for sustainable forest management, carbon accounting, and climate change monitoring [3]. In particular, sustainable forest management contributes to the ecological, economical, and

sociocultural aspects of the environment and requires accurate, consistent, and timely forest monitoring [4].

Large-scale AGB change mapping faces some limitations in both time and space. From the time perspective, there has been a vital demand for AGB mapping and carbon stock monitoring throughout the years [5,6]. On the one hand, traditional methods are destructive, labor-intensive, costly, and time-consuming. On the other hand, covering a large region of interest for national/state-wide AGB mapping using remote sensing techniques requires the processing of big geospatial data. Data collection over time and space raises challenges in big data processing, such as volume, variety, and velocity, which are the characteristics of big data [7–9]. A huge volume of big data comprises issues regarding storage and analysis. Variety is referred as various types and formats of big data. The velocity deals with the unprecedented speed of big geospatial data coming from different sources [9]. Earth observation (EO) data has been extensively used for quantifying forest AGB. While the spectral reflectance of optical and backscatter intensity of synthetic aperture radar (SAR) is used to "indirectly" measure forest vertical structure, lidar, interferometric SAR (InSAR), and photogrammetric 3D products give "direct" measurements of forest vertical structure. Combining direct and/or indirect EO-based forest structure measurements with allometric equations provide information for AGB change monitoring over time and in large areas [10]. Thanks to the freely available remote sensing data of the Landsat mission and cloud computing platforms, such as the Google Earth Engine (GEE) [10], large-scale AGB mapping can be conducted to monitor forest AGB and carbon stock in a timely manner. Recently, cloud computing platforms (e.g., GEE, Amazon Web Services, and Microsoft Azure) paved the road for processing and implementing remote sensing data [9].

Due to the capabilities of GEE for handling geo big data, such as freely available satellite imagery and built-in machine learning models [9,11], the historical AGB mapping over large areas has been facilitated. The AWS and Microsoft Azure platforms also contain machine learning, artificial intelligence services, and several types of satellite imagery. However, GEE provides an access to a comprehensive archive of Landsat imagery since 1972, while AWS and Azure host Landsat 8 OLI datasets [9]. In particular, free aerial imagery, optical, SAR, and ready-to-use data, which are highly important for environmental change monitoring, can be accessed and processed by the GEE cloud platform. Traditional demand to download and store terabytes of satellite imagery in big data analysis can be overcome by GEE infrastructure. The availability of machine learning models in GEE, such as classification and regression tree (CART), random forest (RF), and the support vector machine (SVM), provides great tools to solve different classification and regression problems. Specifically, GEE works based on the parallelizing processes that takes place in the CPUs of Google's data centers, which remarkably reduces remote sensing big data computational time [9]. It should be noted that complex tasks in GEE are required to be paid using the online or batch option. While some might be interested in upgrading their local computers to implement complicated tasks, using a cloud computing platform such as GEE could be a safer choice for the batch processing and storing of a huge volume of data.

In recent years, light detection and ranging (LiDAR), optical, and SAR datasets have been extensively used for forest AGB estimation [12–18]. Landsat data with nearly five decades of continuous imagery, started in 1972, is one of the most valuable sources for historical environmental monitoring [10]. Despite the increasing number of EO satellites with high-spatial and temporal resolutions, Landsat plays a key role in the historical environmental mapping [10]. Landsat provides 30 m spatial and 16 days of temporal resolution, which is of paramount importance for change detection applications, mainly historical AGB mapping [12]. As such, forest AGB mapping using Landsat archive enables researchers to obtain enough information for carbon stock changes and monitor climate change patterns over years. A problem with the indirect measurements of the forest structure and biomass estimation using optical or SAR data is the saturation issue in high-biomass areas. Studies show that backscatter or spectral reflectance (or its derived indices such as NDVI) increases with increasing AGB for low-to-medium AGB, but gradually loses

its sensitivity for higher AGB and asymptotes to a saturation level ([6,17]). LiDAR data, on the other hand, directly measures the forest vertical structure and canopy height and thus are not affected by saturation problems in forest AGB estimation.

AGB mapping using remote sensing can be implemented at two different processing units, pixel-based and object-based image analysis [19]. Object-based image analysis (OBIA) has shown unprecedentedly improved results, especially for AGB mapping [20,21]. This is because extracting textural and morphological (shape, extend, size) features add contextual information to spectral features extracted using pixel-based approaches [19]. Moreover, the mixed pixel issue is solved by applying OBIA through clustering objects based on spectral similarities [19].

The primary objective of this study is to develop an OBIA framework for state-wide historical AGB mapping. So far, some studies have been focused on object-based AGB mapping on a local extent [20–24]. Hirata et al. (2018) [20] used OBIA for AGB estimation in seasonal tropical forests in Cambodia using multiple linear regression and LiDAR data as training. Another object-based study has been done over the mountain tropical Brazilian forest using RF as a machine learning algorithm, terrain, and environmental data as input variables [21]. Chen et al. (2022) [23] have used a combination of spaceborne LiDAR data, optical, and SAR data for estimating the AGB of a protected temperate forest in China using an object-based approach. Since these studies have been conducted over tropical and temperate forests and on a local scale using single-date satellite imagery, this research can contribute to AGB and carbon change monitoring over large areas and more than 20 years. Second, this study aims to use the developed OBIA for the AGB mapping of the entire New York State temperate forests from 2001 to 2019. To achieve this goal, Landsat 5 TM, Landsat 7 ETM, and Landsat 8 OLI images, along with environmental and topographical data, have been processed using the RF regression model in the GEE cloud platform. The developed framework can be transferred to other northeastern states for AGB estimation.

## 2. Study Area and Datasets

### 2.1. Study Area

The study area is the entire New York State, United States, which covers an approximate area of 141,297 km$^2$ (Figure 1). About 61% of New York State is covered by forest, which is about 7.5 million ha of land area across the state [25]. Publicly owned forest land covers at least 1.5 million ha, while privately-owned forest land covers 5.8 million ha, approximately 76% of forest land. More specifically, maple, beech, and birch are the most common forest types, covering 53% of forestland area. New York forestland includes a broad spectrum of species resource in the region, since it contains 94 tree species and 55 forest types [26]. As such, New York's forests play a crucial role in economic resources, wildlife habitat, and recreational opportunities, to name a few.

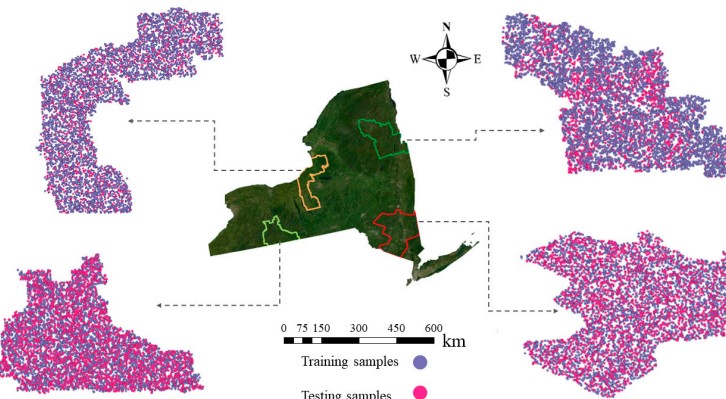

**Figure 1.** Location of the study area (NYS), boundaries of 4 pilot areas of LiDAR data (red: three counties—2014; dark green: Warren Washington Essex—2015; light green: Allegany and Steuben—2016; orange: Cayuga Oswego—2018), training (purple), and validation (pink) samples for each LiDAR raster.

## 2.2. Forest Inventory and Analysis (FIA) Plots

The United States Forest Service (USFS) forest inventory and analysis (FIA) program is the nation's forest census [26]. The FIA program has created a database that can be used to generate a set of reference data consisting of forest inventory information attached to georeferenced plots [27]. These plot data are relatively current, collected in a standardized fashion, and distributed relatively uniformly across the continental United States, Hawaii, US territories, and parts of Alaska [27]. FIA program reports information on the species, size, trees' health, total tree growth, and forest land ownership [26]. Annual FIA inventory plots in NYS are available since 2000, providing a valuable database for training and the precise assessment of AGB maps, especially for large-scale historical monitoring. Thus, this study used these plots to produce and evaluate AGB maps at a state-wide scale. The long history of FIA data provides trend information to resource managers, policymakers, investors, and the public through a system of annual resource inventory that covers both public and private forest lands across the United States [28]. Species-specific allometric equations developed by Jenkins et al. (2003) [29] were used to calculate the AGB of FIA plots.

FIA plots consist of a 0.067 ha (0.17 ac) plot cluster distributed over approximately 0.405 ha (1 ac) [27]. They are collected at a standard sampling intensity of 1 plot per 2428 ha (6000 ac), as in NY and 5198 plots in NY after a full cycle [27]. In each full cycle, which is a 5-year cycle, one fifth of the plots are measured each year, resulting in a full cycle of plots to be completed every 5 years.

The standard FIA plots consist of four circular 24-foot radius (approximately 0.017 ha) subplots and four circular 6.8-foot radius microplots (approximately 0.0013 ha) [30]. Trees 5 inches and greater in diameter were measured in subplots, while trees smaller than 5 inches were measured in microplots. Table 1 lists the statistical characteristics of calculated AGB values of FIA plots from 2002 to 2019.

**Table 1.** The statistical characteristics of AGB values of FIA plots.

| Year | Min_AGB (Mg/ha) | Max_AGB (Mg/ha) | Mean_AGB (Mg/ha) | Median_AGB (Mg/ha) |
|---|---|---|---|---|
| 2002 | 0 | 317.18 | 71.26 | 52.73 |
| 2003 | 0 | 378.73 | 71.81 | 51.22 |
| 2004 | 0 | 298.51 | 67.61 | 37.55 |
| 2005 | 0 | 308.04 | 74.19 | 61.67 |
| 2006 | 0 | 369.94 | 70.80 | 43.69 |
| 2007 | 0 | 318.06 | 71.12 | 54.44 |
| 2008 | 0 | 324.59 | 73.13 | 56.10 |
| 2009 | 0 | 403.79 | 73.73 | 53.59 |
| 2010 | 0 | 320.69 | 73.03 | 49.37 |
| 2011 | 0 | 392.31 | 75.13 | 50.61 |
| 2012 | 0 | 330.86 | 73.34 | 60.18 |
| 2013 | 0 | 327.16 | 76.87 | 61.34 |
| 2014 | 0 | 422.62 | 78.07 | 58.50 |
| 2015 | 0 | 336.47 | 74.10 | 49.20 |
| 2016 | 0 | 360.56 | 79.45 | 62.64 |
| 2017 | 0 | 424.99 | 80.19 | 58.43 |
| 2018 | 0 | 349.01 | 76.99 | 65.12 |
| 2019 | 0 | 322.56 | 79.24 | 57.48 |

*2.3. Remote Sensing Data*

2.3.1. Airborne LiDAR

Figure 1 demonstrates four pilot areas with available airborne LiDAR collected by New York State GIS program office [27] (NYSGPO). The primary purpose of this initial LiDAR data collection was to produce a 1 m cell size high-accuracy 3D digital elevation model (DEM) for conservation planning, design, research, floodplain mapping, dam safety assessments, and elevation modeling [31]. In this study, three counties (i.e., Ulster, Dutchess, and Orange), Warren Washington Essex, Allegany Steuben, and Cayuga Oswego LiDAR datasets were utilized for training/testing purposes, which were collected in 2014, 2015, 2016, and 2018, respectively (Figure 1).

Airborne LiDAR data were collected over four pilot areas in NYS by the New York State GIS program office (NYSGPO) [31]. First, LiDAR data was collected in three counties of Ulster, Dutchess, and Orange for an extent of 7371.11 km$^2$ in 2014 [32] using the Leica ALS70-HP LiDAR system. The nominal pulse spacing for this project was no greater than 0.7 m. The second pilot area, 6278.13 km$^2$, spans five counties, including Warren, Washington, Essex, Hamilton, and Franklin [33], using the ALS70 SN#7123 at a max flying height of 3500 m AGL. This dataset was acquired in 2015, in ground conditions of water at normal levels, no annual inundation, no snow, and leaf off season with nominal pulse spacing of 0.5557 m. LiDAR data for Allegany and Steuben counties, covering approximately 3408.42 km$^2$, was collected in 2016 at a nominal pulse spacing of 0.7 m while no snow was on the ground and rivers were at or below normal levels [34]. Cayuga and Oswego County LiDAR data was collected in 2018 at a nominal pulse spacing of 0.7 m in 2018 [35]. The ground condition of this dataset was at no snow and rivers were at or below normal levels. The data was formatted according to the USNG Tile naming convention, with each tile covering an area of 1500 m by 1500 m.

2.3.2. Landsat Imagery

Since 1972, the Landsat mission, a joint program of the United States Geological Survey (USGS) and the National Aeronautics and Space Administration (NASA), has provided uninterrupted imagery of the Earth's surface [10]. These continuous images are available at a 30 m spatial resolution and 16 days of revisit time [36]. Landsat datasets are produced by USGS in three categories, including Tier1, Tier2, and real-time (RT) [37]. Tier 1 data that meets geometric and radiometric quality requirements were used in this study. As mentioned earlier, this study has been conducted in GEE, where the Landsat satellite images are freely available [9]. First, Landsat images, with less than 5% cloud coverage, were collected for July–August each year. Then a median function was applied to each pixel of the multitemporal set to create the composite image (single image) of the entire NYS for each year. Landsat 5 TM surface reflectance (SR), Landsat 7 ETM+ SR, and Landsat 8 OLI/TIRS SR imagery was used for AGB mapping over the time span of 2000–2011, 2012, and 2013–2020, respectively. To solve the Landsat 7 striping issue, the SLC Gap-Filled Products Phase One Methodology was used. Generally, this technique uses a local linear histogram matching method to fill the scan gap with previously acquired Landsat 7 imagery [38].

First, a Landsat-based detection of trends in disturbance and recovery (LandTrendr) function known as "buildSRcollection" has been used to produce a cloud-masked medoid composite of the Landsat SR TM-equivalent bands [39] for July–August of each year. The output of this step is six spectral bands, including blue, green, red, near-infrared (NIR), shortwave infrared-1 (SWIR1), and shortwave infrared-2 (SWIR2) bands. Then, 13 vegetation indices: normalized difference vegetation index (NDVI), enhanced vegetation index (EVI), ratio vegetation index (RVI), difference vegetation index (DVI), soil adjusted vegetation index (SAVI), normalized green–red difference index (NGRDI), wide dynamic range vegetation index (WDRVI), excess green index (ExG), Chlorophyll Index–green (CI green), visible atmospherically resistant index (vari), green leaf index (GLI), normalized burn ration (NBR), normalized difference moisture index (NDMI), and four tasseled cap

coefficients: brightness, wetness, greenness, and angle were extracted using spectral bands. LandTrendr-GEE code [40] was used to extract LandTrendr metrics, such as fit-to-vertex (FTV) data (i.e., LandTrendr fitted brightness, LandTrendr fitted greenness, LandTrendr fitted wetness, and LandTrendr fitted normalized burn ratio) corresponding to LiDAR data year, delta matrices (difference between each year's FTV and the previous year), magnitude, and disturbance matrices since last disturbance.

### 2.4. Climate and Topographic Data

Climate and topographic datasets were also included as predictors in the RF model for historical AGB mapping. Climate data were downloaded from the parameter-elevation regressions on independent slopes model (PRISNM) climate group at Oregon State University [41] at 30-year normals. The PRISM data are models while the term "Normals" refers to them being 30 year averages. These datasets cover the period 1981–2020, which contains the average annual conditions over the most recent three full decades [42]. For this study, mean annual precipitation, minimum annual temperature, and maximum annual temperature were used in addition to other input variables. Moreover, a 30 m topographic wetness index (TWI) surface was also added to predictors. The topographic data originated from a 30 m DEM downloaded using the terrainr [43] package, which provides ease-of-use functions for accessing DEMs from the United States Geological Survey National Map [44]. Slope and upslope were also extracted from DEM. Both slope and upslope area were computed from the original 30 m DEM. TWI was then calculated as follows:

$$TWI = \log\left(\frac{\text{upslope area}}{\tan(\text{slope} + 0.0001)}\right) \tag{1}$$

Note that 0.0001 was added to the slope to prevent a 0 in the denominator.

DEM, aspect, and slope were included in input predictors as topographic data. These datasets were downloaded from Shuttle Radar Topography Mission (SRTM) products at 30 m resolution.

## 3. Methods

Our state-wide AGB mapping approach comprises three main steps, as depicted in Figure 2. First, airborne LiDAR datasets (Section 2.1) were used to produce LiDAR AGB rasters for each pilot area for 2014, 2015, 2016, and 2018. Second, Landsat imagery were preprocessed (e.g., cloud masking) in GEE to extract spectral bands and vegetation indices. In addition, Landsat-based detection of trends in disturbance and recovery (LandTrendr) algorithm [45,46] has been applied to compute LandTrendr metrics. Thus, spectral bands, vegetation indices, LandTrendr, climate, and topographical data were stacked as input predictors for further analysis. In the next step, the segmentation has been applied using the simple noniterative clustering (SNIC) technique in GEE to extract objects for OBIA. Then, stratified random sampling of generated LiDAR AGB rasters was used to split the reference data for training/testing purposes. Since pilot areas belong to different years, Landsat data for each year have been used separately. For each pilot area, 6000 objects were used as reference samples. Then, all samples for all pilot areas were merged to train the RF model for the whole NYS. Finally, a stacked layer of input predictors for each year (2001–2019) was used to provide the historical AGB maps using the OBIA approach. For historical state-wide AGB mapping, a LiDAR-based training method, which was developed by Hudak et al. (2020) [12], has been employed to produce the historical AGB maps of the entire NYS from 2001 to 2019.

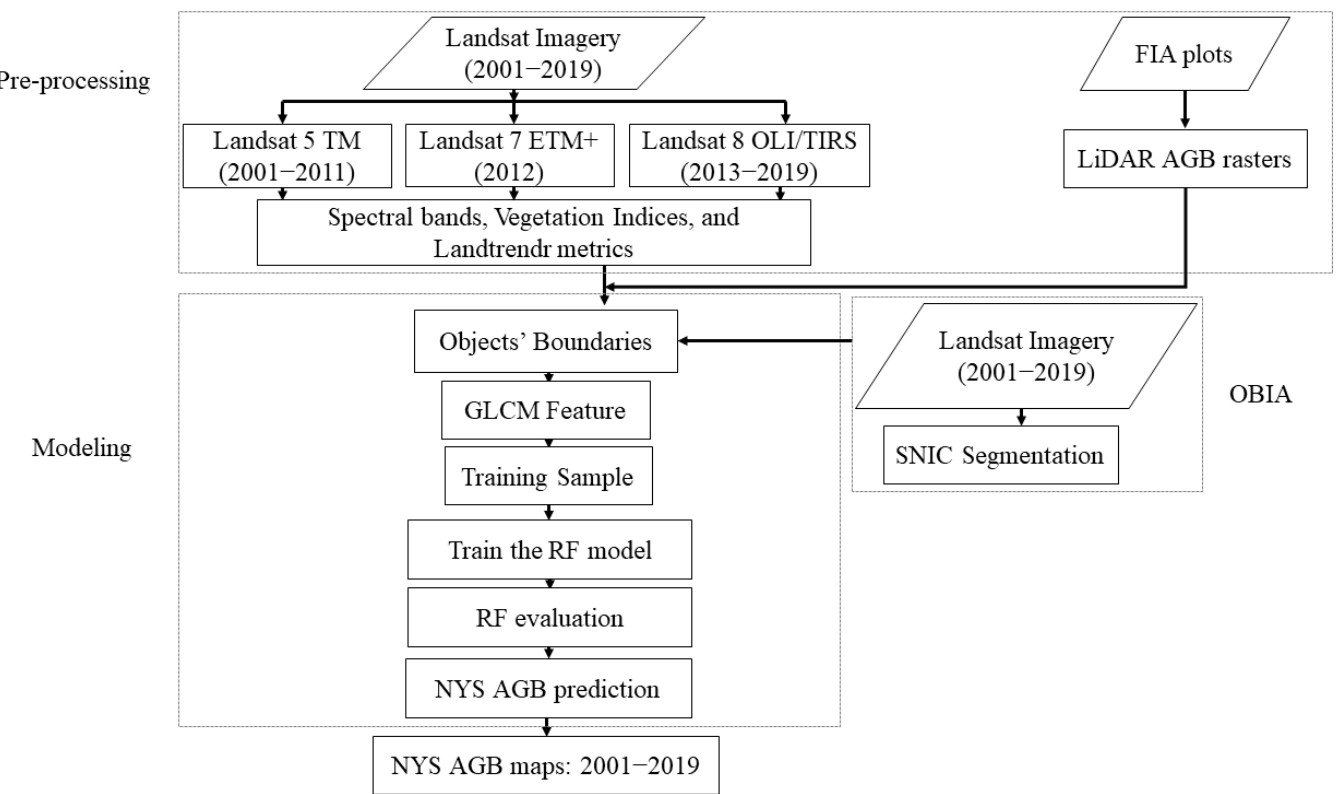

**Figure 2.** The proposed object-based AGB estimation workflow using random forest regression model and Landsat imagery.

### 3.1. Airborne LiDAR AGB Raster

One of the challenges in applying machine learning models is the need for enough training samples. Having access to private field measurements, such as FIA plots, is very limited due to the legal concerns. In addition, even with access to FIA plots, their sampling density is relatively sparse both in space and time (repetition in each year) [47]. From the spacing perspective, there is only one plot for each 25 square kilometers. From the time perspective, not all plots are measured each year. In fact, the legislative mandates require measurements of 10% to 20% of plots in each State, each year [47]. To leverage training samples, several studies tried to produce AGB rasters using airborne LiDAR predictors and use these airborne LiDAR-derived AGB rasters as a reference dataset [12,20]. We adapted this approach as described in Hudack et al. (2020). We first generated airborne LiDAR AGB for 4 pilot areas (Figure 1) using an RF model. We then used a stratified random sampling method to create a training/testing split. For stratified random sampling, samples were sorted from 0 to maximum AGB with 2 Mg/habins and 200 objects were randomly extracted from each bin. For OBIA, objects' boundaries were overlaid on airborne LiDAR-produced AGB maps to calculate the mean AGB within each object. When a bin had less than 200 objects, one-half of the objects were randomly selected [12].

The procedure of generating airborne LiDAR AGB rasters started with row-wise observation filtering, deleting observations that indicate poor alignment between FIA and airborne LiDAR data at a plot. For instance, at this stage, observations with missing values, plots with 0 Mg/ha AGB and a maximum LiDAR height above 10 m, and plots with 500 Mg/ha AGB and a maximum LiDAR height below 10 m were deleted. Second, a set of predictors containing airborne LiDAR height, intensity, environmental, topographical, and tax parcel codes (cadastral information related to land-use and management) was created to train the model. Then, the final set of predictors was split into a training set (containing 70% of the data) and a holdout test set (containing the other 30% of the data). It is worth mentioning that the training set is used for all model tuning and validation stages, while

the test set is used to assess the performance of the final model. Finally, the ranger package in R was used to implement an RF regression model to produce airborne LiDAR AGB rasters for each pilot area. A grid search approach was used to determine values for model hyperparameters. For each combination of values, model performance was assessed using the training set and 5-fold cross validation. Parameter combinations are then ranked by mean RMSE. The best performing ranges of each parameter are then used to build another grid, this time using a more constrained set of values. This process is repeated until a relatively steady set of parameters are identified; while there is no strict stopping rule used, the best models at the final tuning step are typically separated by less than 1% RMSE. It is certainly a subjective process. However, we tuned until there was no point in refining the hyper parameter values any further. Here is an example with the "mtry" parameter: First, tune: values 1–5, winner is 5. Second, tune: values 1–10, winner is 10. Third, tune: values 8–15, winner is 13. Stop there. We have tuned be smallest increment (1, it does not make sense to tune mtry by fractions) and found the winner. This implementation provides the following hyperparameters to tune:

- num.trees: The number of trees to aggregate. Values evaluated span 50–5000.
- mtry: Number of variables to split at in each node. The initial grid evaluates using 5–40% of variables at each split (for our current dataset, 3–40); our end model typically uses 40–50% of variables at each split.
- min.node.size: Minimum node size (number of observations in the terminal nodes of each tree).

    - Higher values result in less complex trees (which can reduce overfitting with noisy predictors). Our initial grid evaluates values from 1 to 10, while our end model typically uses a value around 6 (with the default for regression being 5).

- replace: Boolean: Take samples with replacement? Tends to be TRUE.
- sample.fraction: What fraction of observations to sample. Our initial grid evaluates values from

    - 0.33 to 0.8; this parameter tends towards either 0.2 or 0.8.

- splitrule: Whether to use maximally selected rank statistics (maxstat) or estimated response variances (variance) as a variable selection splitting rule.

It should be noted that in the process of generating airborne LiDAR data, inevitably, errors are included in the final predicted model. Thus, although airborne LiDAR AGB rasters provide sufficient training samples, the potential propagation of errors from these reference datasets into our model predictions should be considered as well.

### 3.2. Object-Based Feature Extraction

An object-based approach has shown promising results in image analysis and image classification applications [19]. However, its potential has not been explored completely in forest AGB estimation. Thus, in this study, the OBIA technique, along with the RF regression model, was used to investigate the ability of OBIA for historical forest AGB mapping. OBIA approach initially segments pixels into objects based on their spectral similarity [48]. OBIA mitigates the "mixed pixels" issue existing in a pixel-based approach by categorizing similar pixels (canopy covers) into objects [19]. Among various segmentation techniques, the simple noniterative clustering (SNIC) algorithm in GEE [48] was used to cluster pixels of Landsat imagery in each year. The SNIC algorithm works by initializing centroids (superpixels) with pixels chosen in the image [49]. It selects the pixels one-by-one from the queue and assigns them to a superpixel until no more pixels remain unassigned. The next pixel to be assigned to a superpixel is chosen by selecting the pixel with the lowest metric value to any superpixel [50]. There are some parameters such as size, compactness, connectivity, neighborhoodSize, and seeds that need to be determined. In this research, a trial-and-error approach was taken to select the best value for each parameter. It is worth mentioning that segments vary through time because biomass changes by the time. First, SNIC segmentation was applied to Landsat imagery for each year separately to create the

objects. The values were selected based on the shape of the objects and the RMSE of the RF model in an iterative process. In each iteration, training/testing samples from LiDAR AGB rasters were calculated based on produced objects using different parameter values. Then, the RF model was run to evaluate the performance of the model using the RMSE. The best RMSE and objects' shape were considered to choose the wining values for SNIC segmentation. Finalized object boundaries were used to generate training/testing samples from LiDAR AGB rasters. Finally, mean, variance, angular second moment (ASM), contrast, entropy, and homogeneity, known as gray level co-occurrence matrix (GLCM) features, within each object have been calculated to create the input set of predictors.

Figure 3 shows a true color composite of Landsat 5 imagery and the result of segmentation by applying the SNIC algorithm using different values for seeds. According to Figure 3b,c, changing the seeds value has changed the object boundaries and segments in (b) are closer to the boundaries of objects on the ground.

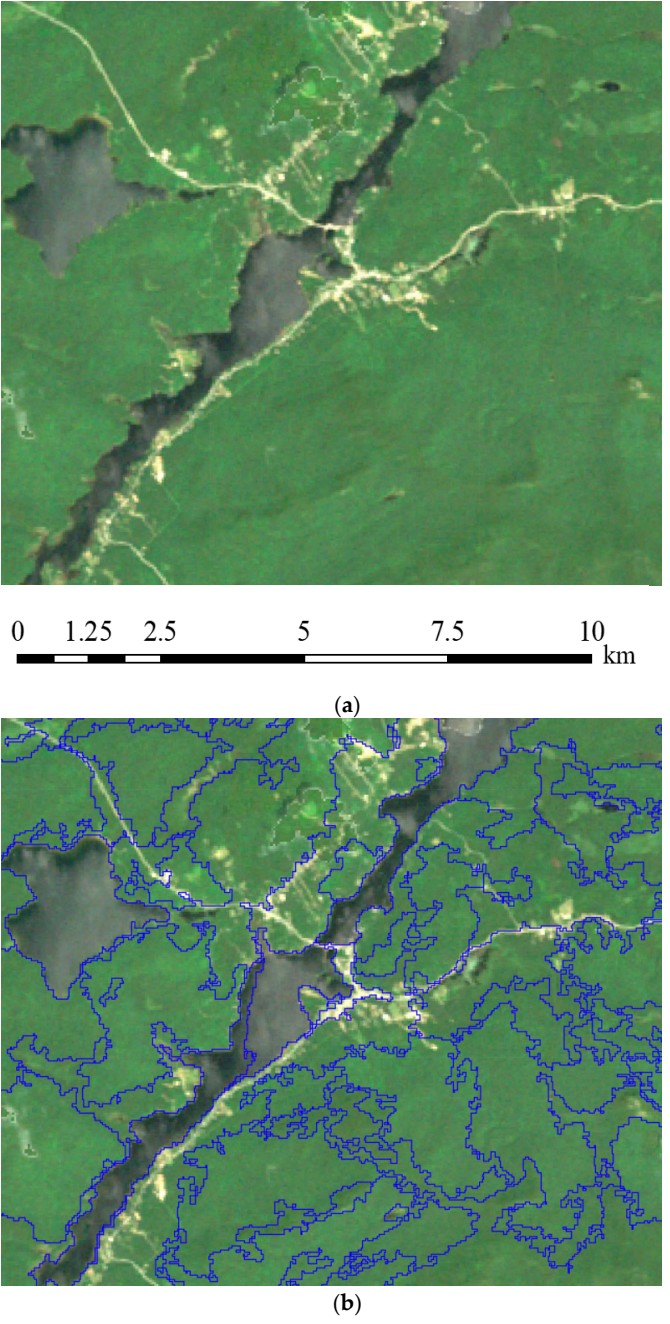

(a)

(b)

**Figure 3.** *Cont.*

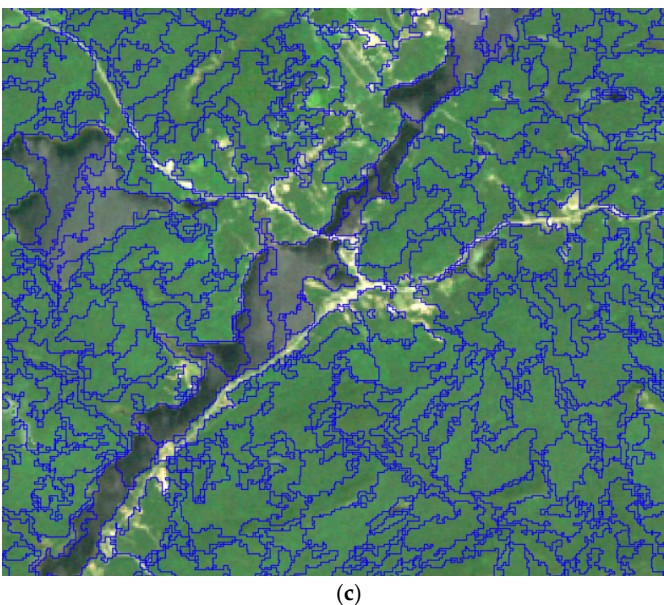

(**c**)

**Figure 3.** An example of SNIC segmentation in GEE using different seeds and same size, compactness, connectivity, and neighborhoodSize. (**a**) Landsat 5 image collection for July/01/2011 to August/31/2011. (**b**) SNIC segmentation using 60 seeds. (**c**) SNIC segmentation using 20 seeds.

### 3.3. Map Accuracy Assessment Using FIA Plots

Because of the limitations in type, magnitude, frequency, and the location of errors that exist in geospatial data assessments, and due to the uncertainties regarding the spatial support between modeled and reference datasets, another approach is suggested to be utilized here. The adopted map accuracy assessment protocol was developed by [27]. It has the potential to overcome the mentioned limitations. Additionally, the uncertainties about spatial support between predicted and measured datasets were considered through accounting for the mismatches in spatial support between the two datasets. First, their approach overcomes the limitations in type, magnitude, frequency, and the location of errors. Second, they consider mismatches in spatial support between modeled and reference datasets, and account for their uncertainties. Although this approach is fundamentally an individual pixel-based comparison between AGB predictions and FIA plot estimates, it can be used for validating object-based predictions. It should be noted that the smallest sample unit of OBIA is inherently a pixel. In an object-based approach, we are dealing with pixels grouped as an object by their reflectance similarity. Thus, it is highly recommended to use sensors with high spatial resolutions for generating object boundaries.

This protocol consists of four types of assessments that incorporate both graphical and geographical visualizations and both qualitative and quantitative measures of agreement.

Assessment 1—Examining data distribution (at several scales):

An empirical cumulative distribution function (ecdf) is a valuable tool for comparing the distributions of modeled and reference datasets qualitatively. It can identify the differences in data distribution such as 0's, max values, and missing range of values. The Kolmogorov–Smirnov (KS) statistic quantifies the agreement between two dataset's distribution in terms of the maximum difference in their empirical distribution [27]. The KS statistic is defined as:

$$D_{KS} = \max |F(x) - G(x)| \tag{2}$$

where $F(x)$ and $G(x)$ are the cumulative distribution functions of reference data and the estimated values.

Assessment 2—Examining overall agreement of area estimates:

In this scenario, a comparison of modeled estimates to measurements from reference datasets is conducted using several types of relative errors. A scatter plot of modeled vs. plotted biomass is used to find systematic or bias and unsystematic or random differences.

The geometric mean functional relationship (GMFR) regression line is a symmetric regression model that can be used to describe the relationship between modeled and reference datasets.

We used several agreement statistics representing systematic and random differences between the two datasets (i.e., model and FIA plots). Most of these statistics are explained in Riemann et al. (2010). The statistics include R2 value, root mean square error (RMSE), mean bias error (MBE), and mean absolute error (MAE). In addition, the agreement coefficient (AC), developed by [51], which is a systematic, symmetric, bounded, and a nondimensional measure of agreement and is defined as follows:

$$AC = 1 - \frac{SSD}{SPOD} \tag{3}$$

SSD is:

$$SSD = \sum_{i=1}^{n} (X_i - Y_i)^2 \tag{4}$$

SPOD is:

$$SPOD = \sum_{i=1}^{n} (|\underline{X} - \underline{Y}| + |X_i - \underline{X}|)(|\underline{X} - \underline{Y}| + |Y_i - \underline{Y}|) \tag{5}$$

where $\underline{X}$ and $\underline{Y}$ are the mean values of datasets $X$ and $Y$, respectively.

The AC value ranges $\leq 0 - 1$, where AC = 1 means the agreement between $X$ and $Y$ is perfect, while values less than or equal to zero is a no agreement indicator.

Ji and Gallo (2006) described an unsystematic sum of product difference (SPD$_u$), which is defined as:

$$SPD_u = \sum_{i=1}^{n} (|X_i - \hat{X}_i|)(|Y_i - \hat{Y}_i|) \tag{6}$$

where $\hat{X}$ and $\hat{Y}$ are from the GMFR regression model and SPD$_s$ can be obtained by SPD$_s$ = SSD − SPD$_u$. Thus, systematic and unsystematic agreement coefficients can be calculated by:

$$AC_{sys} = 1 - \frac{SPD_s}{SPOD} \tag{7}$$

$$AC_{uns} = 1 - \frac{SPD_u}{SPOD} \tag{8}$$

where AC$_{sys}$ = 1 if the GMFR line is perfectly in line with the 1:1 line and AC$_{uns}$ = 1 if all points fall directly on the GMFR line.

Assessment 3—Examining spatial and distribution pattern of local differences:

Due to the spatial variations of field measurements in ground condition, quality, and the local applicability of regression model techniques, the predicted values may have different levels of error across the area of interest. Accuracy assessment throughout the region of interest will provide important information regarding the magnitude and direction of the relative error. To conduct this assessment, choosing the optimal size of the scale is of paramount importance. The scale needs to be to be large enough to contain sufficient reference data for associated confidence intervals to be of an acceptable size. In parallel, it has to be small enough to provide a reasonable spatial illustration of regional variation of study area.

Assessment 4—Examining local variability:

Optimizing the accuracy for local or global processes, a lack of input information at the desired scale, etc., can lead to loss of local variability. A choropleth map of local variance or standard deviation presents a qualitative interpretation of the magnitude and spatial patterns of differences between datasets in the study area. Local variability provides information on local spatial variability in the modeled dataset and its relationship to the reference dataset.

The following lines describe the FIA plot inclusion criteria:

All subplots must be sampled.

Annual assessment vs. pooled assessment:

For an annual assessment: only FIA plots sampled in the designated year were considered and compared to the modeled AGB surface for said year.

For a pooled assessment: FIA plots inventoried in all years (2001–2019) were aggregated within each hex. Modeled values were extracted from each annual surface to match the inventory year for a particular FIA plot but aggregated within each hex just the same.

## 4. Results and Discussion

The following subsections describe the qualitative and quantitative evaluation of the produced AGB maps.

### 4.1. Historical AGB Mapping

The national land cover database (NCLD) only provided the land cover maps for 2001, 2006, 2011, 2016, and 2019, respectively; hence, in order to analyze the change over the last 19 years, these available maps were used. Figure 4 shows the AGB maps of the entire NYS for 2001, 2006, 2011, 2016, and 2019. As the figure shows, the AGB has increased in the central upstate part of the NYS between 2001 and 2006, followed by regrowth in the northern regions in 2011. According to the 2016 AGB map, the AGB has increased in comparison to 2011. However, there is no remarkable AGB change between 2016 and 2019.

Loss, gain, and unchanged areas are demonstrated in Figure 5 for the 5-year increment. Areas with gain are shown in green, loss in red, and unchanged regions in blue. For the 2001–2006 time period, green areas are more than red areas, which indicates an increase in the amount of AGB in the year 2006. In the 2006–2011 change map, the loss seems to be the dominant phenomenon (Figure 5). For the 2011–2016 AGB change map, both loss and gain are evident. Within the time span of 2016–2019, it is hard to see significant loss or gain. In order to analyze further the loss/gain of AGB, specific regions across NYS were selected (Figures 6 and 7) and investigated according to the US National Maps Attributing Forest Change: 1986–2010 [45].

Schleeweis et al. (2020) [45] utilized the Landsat archive and an ensemble of disturbance algorithms to produce the US national maps attributing forest change from 1986 to 2010. The disturbance map includes the following classes: stable (i.e., no change), removals, fire, stress, wind, conversion, and other. The most dominant class of forest change in NYS is removal. For all training datasets, land cover and land use observations were collected. Plots labeled as mechanical forest change were labeled as removal or conversion depending on the evidence of vegetation recovery and/or land use change. The removal class has 82.3% user's and 72.2% producer's accuracy, respectively. Figure 6 shows the removal disturbance maps of a region in north part of NYS from 2000 to 2010. The AGB map of each year clearly shows the removal occurrence according to that year, which indicates the reliability of our RF historical AGB modeling.

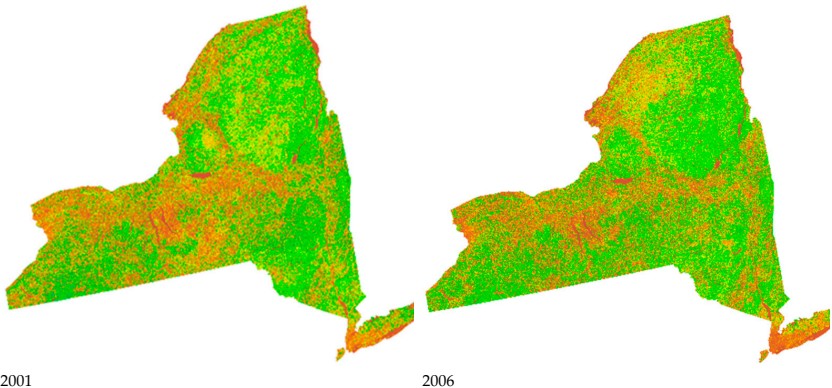

2001 2006

**Figure 4.** *Cont.*

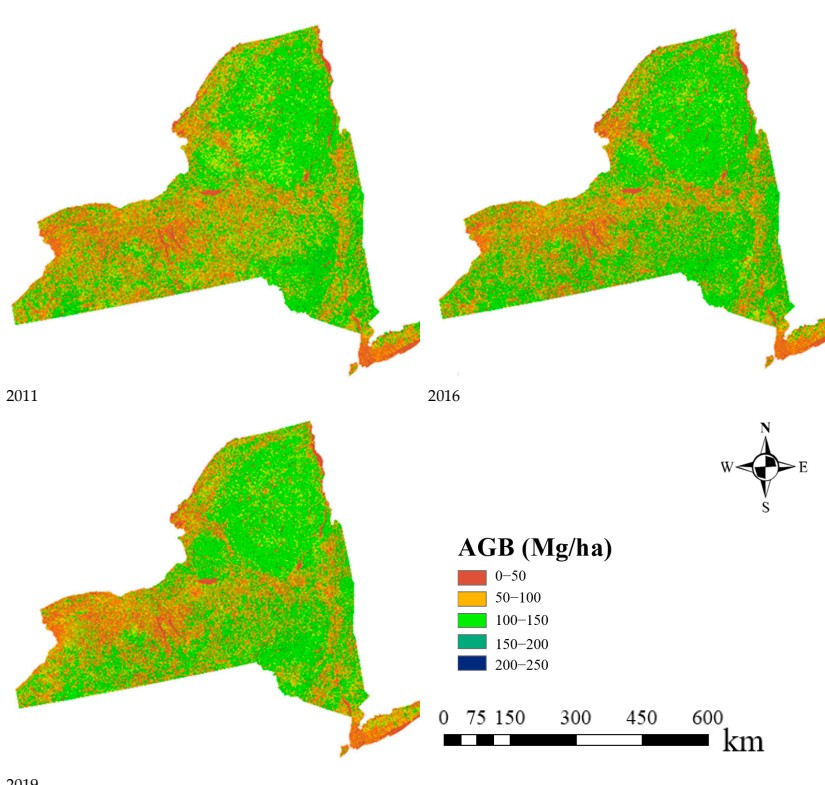

**Figure 4.** NYS object-based historical AGB mapping using random forest regression model in GEE and Landsat imagery.

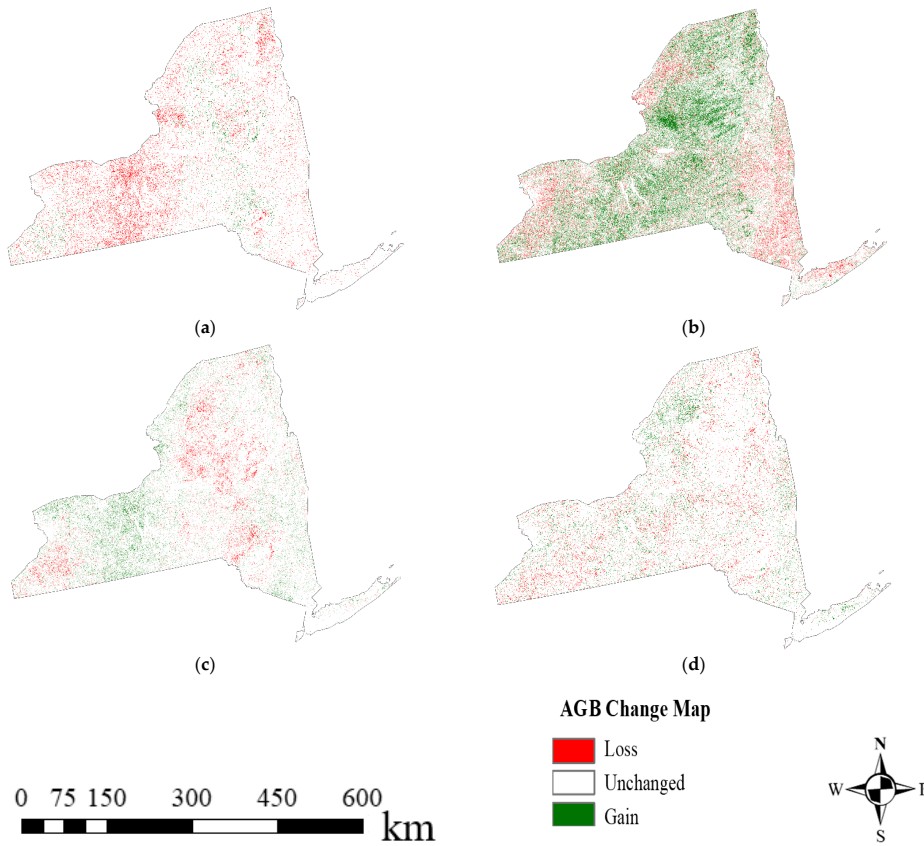

**Figure 5.** The 5-year increment change maps to analyze loss (red), gain (green), and unchanged (blue) areas in NYS for time spans: (**a**): 2001–2006, (**b**): 2006–2011, (**c**): 2011–2016, and (**d**): 2016–2019.

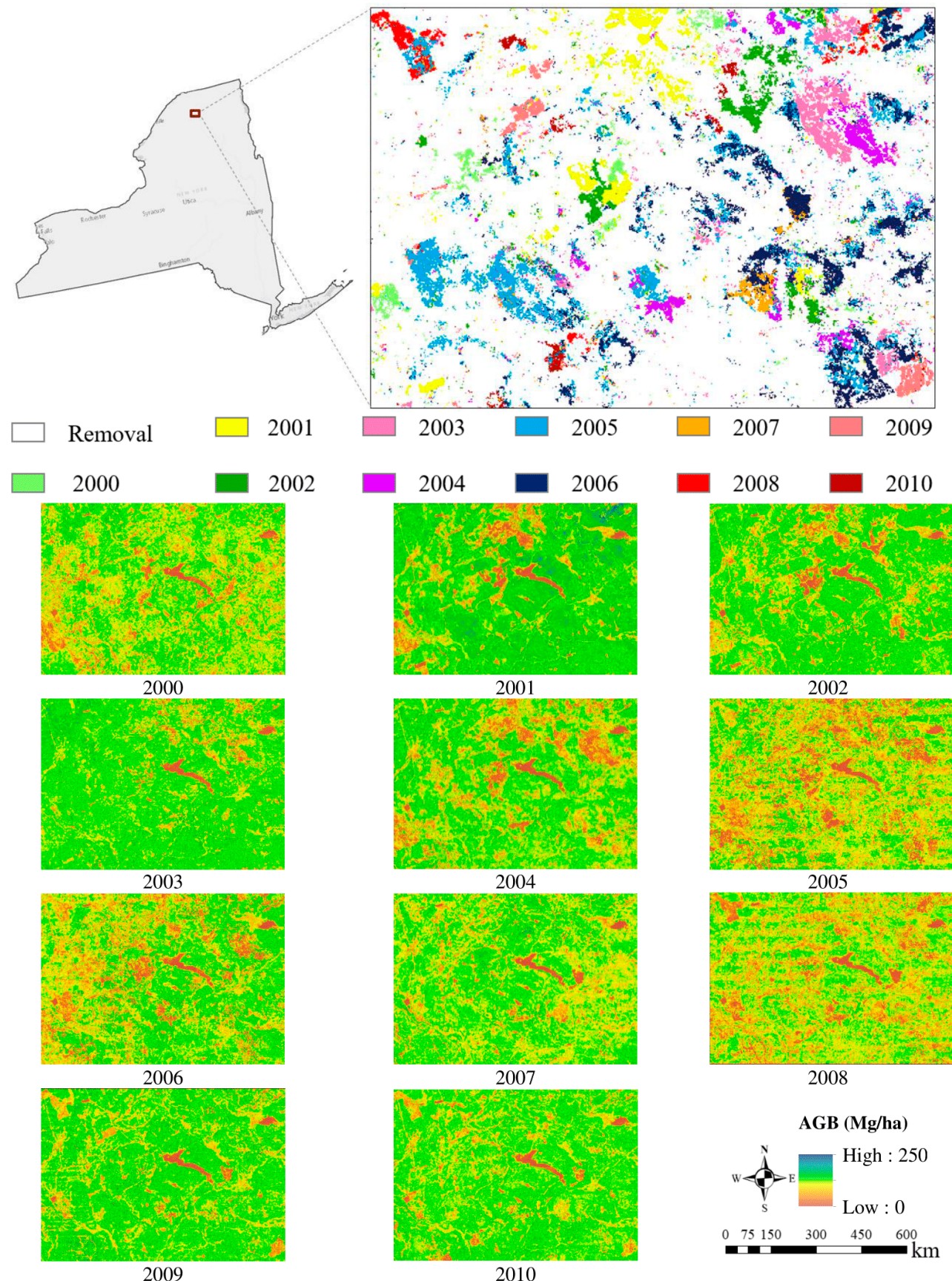

**Figure 6.** Forest change (removal) from 2000 to 2010 in an area in northern NYS and the related AGB maps.

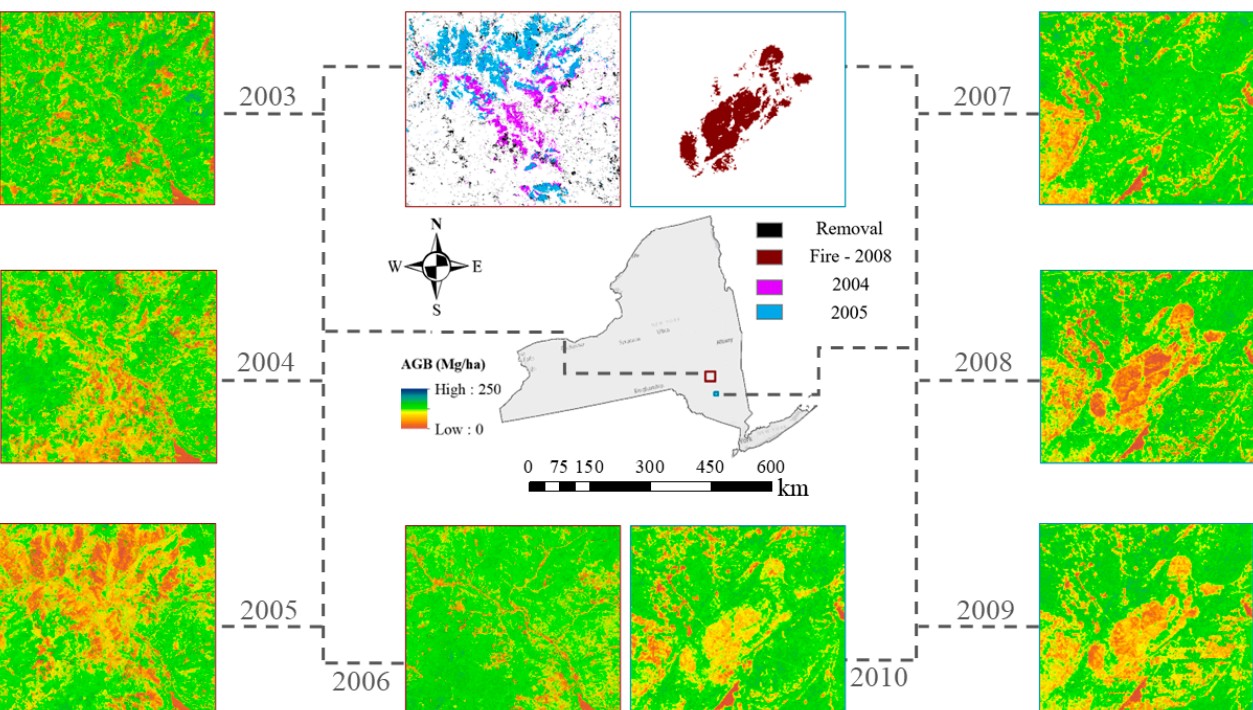

**Figure 7.** Two zoomed areas for demonstrating the forest disturbance (left: forest removal in 2004 and 2005; right: forest fire in 2008).

In addition, Figure 7 shows two zoomed areas of forest disturbance, with one demonstrating the removal occurred in 2004 and 2005, while the other indicates the forest fire in 2008. First, the rectangles on the left show the AGM map of the years from 2003 to 2006. The removal disturbance has been clearly demonstrated in the AGB maps of 2004 and 2005, respectively. Second, according to Figure 7, the 2007 AGB map is before the fire and the fired area is green. Then, in 2008, the year of forest fire, the map clearly shows the burned region with red area indicating close to zero AGB values. Importantly, the AGB maps of 2009 and 2010 show the recovery/regrowth of forest in this region.

Changes in the forest and three different forest types have been calculated based on the NLCD maps classification. Changes in the forest and three different forest types have been calculated based on the NLCD maps classification. The NLCD map consists of a 30 m resolution land cover classes with a 16-class legend based on a modified Anderson Level II classification system [52]. First, the raster of each forest class type, including deciduous, evergreen, and mixed forests, was converted to a shapefile. Second, the shapefiles were used to clip the AGB maps to calculate the amount of the AGB for each year. Then, the AGB changes were calculated using the difference of AGB values between years. Figure 8 shows the total forest AGB of 2001, 2006, 2011, 2016, and 2019. As shown, there is a decrease in the amount of AGB from 2001 to 2006. Then, there was a considerable increase in 2011, followed by a slight increase in 2016. The AGB's loss from 2016 to 2019 can be seen in Figure 8. Figure 9 demonstrates the AGB changes in $10^6$ Mg/ha for 2001, 2006, 2011, 2016, and 2019 in deciduous, evergreen, and mixed forests using NLCD maps. First, there is a decrease, $983.8 \times 10^6$ Mg/ha, in deciduous forests, about $132.7 \times 10^6$ Mg/ha for evergreen forests, and $173.2 \times 10^6$ Mg/ha for mixed forests from 2001 to 2006. From 2006 to 2011, deciduous, evergreen, and mixed forests areas have increased $618.3 \times 10^6$ Mg/ha, $82.5 \times 10^6$ Mg/ha, and $137.4 \times 10^6$ Mg/ha, respectively. Again in 2016, a regrowth can be seen in deciduous, evergreen, and mixed forests in comparison to 2011 for about $229.1 \times 10^6$ Mg/ha, $23.5 \times 10^6$ Mg/ha, and $27.8 \times 10^6$ Mg/ha, respectively. Deciduous, evergreen, and mixed forests have decreased $67.2 \times 10^6$ Mg/ha, $41.9 \times 10^6$ Mg/ha, and $26.4 \times 10^6$ Mg/ha, respectively, in 2019.

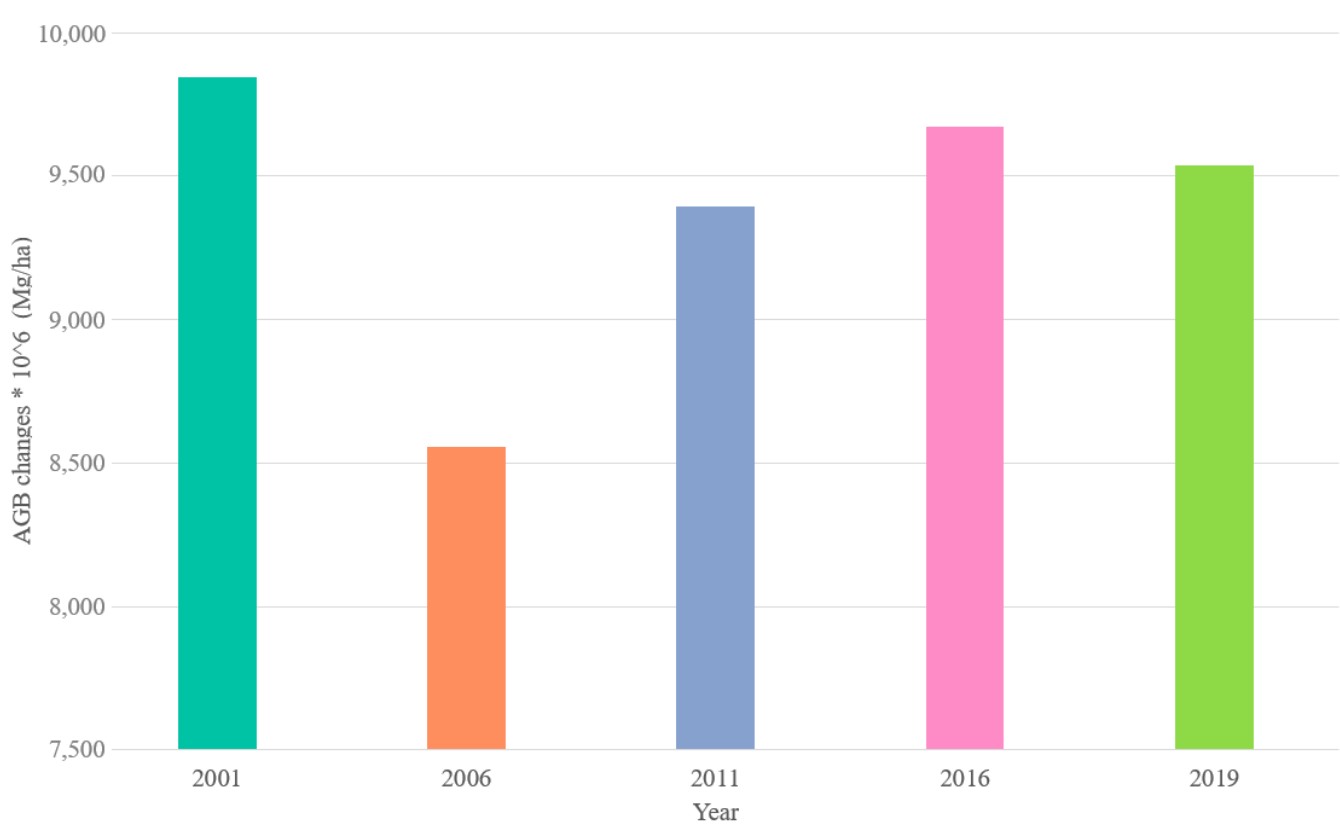

**Figure 8.** Total forest AGB changes over time in NYS (2001, 2006, 2011, 2016, and 2019).

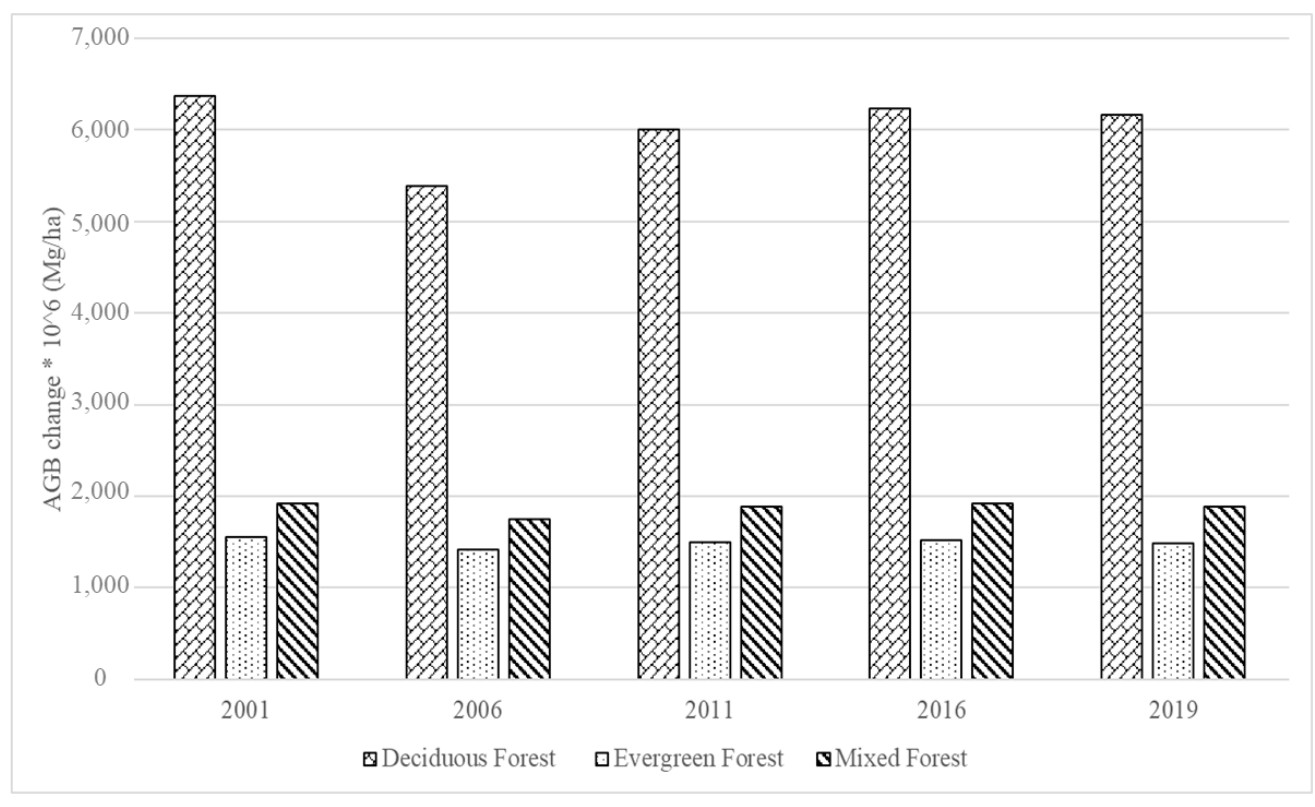

**Figure 9.** Forest type changes over time in NYS using classes of NLCD maps (2001, 2006, 2011, 2016, and 2019).

### 4.2. Accuracy Assessment Using FIA Plots

Results of Assessment 1: The KS distance measures the maximum difference between the ecdf of the model and that of the FIA plot data. In Tables 2–5, the KS distance is the largest value at the plot:pixel scale for all years because a much larger number of plots with very small biomass values are included. For the years 2019 and 2016, the KS distance at 78,100 ha and 216,500 ha scales is decreased close to zero because the two distributions became more similar (Tables 4 and 5, and Figure 10).

**Table 2.** Agreement statistics (SI-corrected)—pixel comparison.

| Group | *n* | PPH | Mean FIA | MBE (Mg/ha) | RMSE (Mg/ha) | MAE (Mg/ha) | $R^2$ | KS | AC | ACs | ACu |
|---|---|---|---|---|---|---|---|---|---|---|---|
| target_2002 | 1017 | NA | 71.33 | 18.53 | 52.71 | 41.83 | 0.56 | 0.35 | 0.49 | 0.84 | 0.65 |
| target_2006 | 880 | NA | 70.88 | 9.47 | 58.66 | 46.04 | 0.47 | 0.36 | 0.15 | 0.71 | 0.43 |
| target_2011 | 940 | NA | 75.21 | 12.03 | 56.66 | 45.14 | 0.55 | 0.34 | 0.32 | 0.75 | 0.57 |
| target_2016 | 640 | NA | 79.45 | 8.03 | 53.13 | 41.25 | 0.59 | 0.34 | 0.39 | 0.81 | 0.57 |
| target_2019 | 606 | NA | 79.46 | 5.76 | 54.53 | 42.67 | 0.60 | 0.38 | 0.30 | 0.75 | 0.56 |
| pooled | 14,333 | NA | 74.24 | 14.09 | 56.22 | 44.55 | 0.53 | 0.35 | 0.34 | 0.77 | 0.57 |

**Table 3.** Agreement statistics (SI-corrected)—8660 Ha Hex.

| Group | *n* | PPH | Mean FIA | MBE (Mg/ha) | RMSE (Mg/ha) | MAE (Mg/ha) | $R^2$ | KS | AC | ACs | ACu |
|---|---|---|---|---|---|---|---|---|---|---|---|
| target_2002 | 921 | 1.10 | 71.75 | 18.84 | 51.90 | 41.03 | 0.56 | 0.32 | 0.49 | 0.84 | 0.65 |
| target_2006 | 821 | 1.07 | 72.20 | 8.91 | 57.69 | 45.30 | 0.47 | 0.34 | 0.13 | 0.70 | 0.43 |
| target_2011 | 855 | 1.10 | 76.67 | 11.62 | 56.06 | 44.62 | 0.55 | 0.32 | 0.30 | 0.73 | 0.57 |
| target_2016 | 576 | 1.11 | 78.49 | 8.66 | 52.0 | 40.07 | 0.60 | 0.34 | 0.41 | 0.82 | 0.59 |
| target_2019 | 526 | 1.15 | 80.30 | 5.17 | 53.64 | 40.89 | 0.60 | 0.35 | 0.28 | 0.73 | 0.54 |
| pooled | 1528 | 9.37 | 73.58 | 14.38 | 36.67 | 29.84 | 0.66 | 0.23 | 0.54 | 0.78 | 0.76 |

**Table 4.** Agreement statistics (SI-corrected)—78,100 Ha Hex.

| Group | *n* | PPH | Mean FIA | MBE (Mg/ha) | RMSE (Mg/ha) | MAE (Mg/ha) | $R^2$ | KS | AC | ACs | ACu |
|---|---|---|---|---|---|---|---|---|---|---|---|
| target_2002 | 193 | 5.25 | 65.47 | 20.25 | 32.37 | 27.31 | 0.71 | 0.25 | 0.67 | 0.82 | 0.84 |
| target_2006 | 191 | 4.60 | 69.59 | 10.10 | 36.15 | 29.77 | 0.60 | 0.26 | 0.37 | 0.70 | 0.67 |
| target_2011 | 190 | 4.94 | 74.31 | 12.00 | 32.76 | 27.02 | 0.73 | 0.23 | 0.57 | 0.77 | 0.80 |
| target_2016 | 184 | 3.48 | 78.36 | 6.64 | 38.17 | 28.00 | 0.68 | 0.20 | 0.46 | 0.78 | 0.68 |
| target_2019 | 179 | 3.37 | 79.99 | 5.29 | 37.20 | 28.44 | 0.65 | 0.17 | 0.41 | 0.78 | 0.63 |
| pooled | 201 | 70.16 | 69.96 | 15.76 | 25.81 | 21.12 | 0.81 | 0.27 | 0.66 | 0.76 | 0.90 |

**Table 5.** Agreement statistics (SI-corrected)—216,500 Ha Hex.

| Group | *n* | PPH | Mean FIA | MBE (Mg/ha) | RMSE (Mg/ha) | MAE (Mg/ha) | $R^2$ | KS | AC | ACs | ACu |
|---|---|---|---|---|---|---|---|---|---|---|---|
| target_2002 | 81 | 12.44 | 62.62 | 22.44 | 30.95 | 25.71 | 0.76 | 0.32 | 0.66 | 0.77 | 0.90 |
| target_2006 | 79 | 10.96 | 69.56 | 10.89 | 33.11 | 26.47 | 0.47 | 0.33 | 0.24 | 0.65 | 0.59 |
| target_2011 | 79 | 11.71 | 73.07 | 14.09 | 31.90 | 22.84 | 0.55 | 0.24 | 0.52 | 0.80 | 0.72 |
| target_2016 | 77 | 8.31 | 75.93 | 8.69 | 26.79 | 21.04 | 0.78 | 0.18 | 0.68 | 0.86 | 0.82 |
| target_2019 | 74 | 8.09 | 79.47 | 5.38 | 30.79 | 24.71 | 0.64 | 0.27 | 0.40 | 0.77 | 0.63 |
| pooled | 82 | 169.79 | 68.75 | 17.81 | 25.87 | 21.20 | 0.81 | 0.33 | 0.62 | 0.71 | 0.92 |

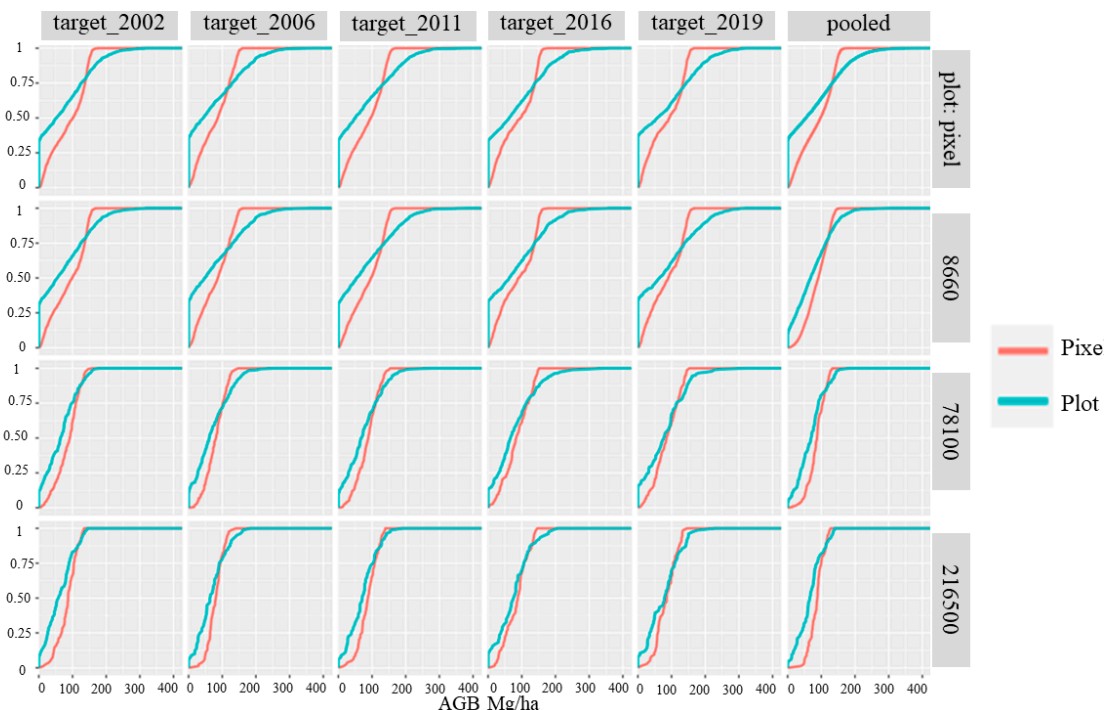

**Figure 10.** ECDF comparisons across scales—FIA plots vs. mapped (SI-corrected).

Although increasing the scale may be associated with the decrease in the KS distance, this decrease may be due to a difference in spatial support and/or dataset uncertainty than the actual difference between the modeled and reference dataset.

Results of Assessment 2: Table 2 also presents the $R^2$ and RMSE values for each dataset and scale. Plots per hexagons (PPH) is computed as the average number of plots in each hexagon for a given aggregation size. Our results support those found in Ji and Gallo (2006) that the $R^2$ value does not reflect the level of systematic difference present in the relationship and simply increases with increasing scale. The RMSE values reflect the effects systematic and unsystematic differences and provide valuable information regarding differences between the datasets in data units. Based on RMSEs only, it is difficult to establish what portion of the difference is due to systematic versus unsystematic disagreement. Increasing the scale also indicates improvements in the RMSE and $R^2$ patterns, which might be due to considering different reference samples at larger scales.

Figure 10 shows the ecdf graph comparisons over the years (i.e., 2002, 2006, 2011, 2016, and 2019) with different scales. Initially, there is no flat section at the beginning of the pixel ecdf for all graphs, which indicates that the generated AGB maps are not able to predict as many as zero AGB values as the FIA data. This means that the NYS AGB maps include less mapped zeros for developed and water areas, expected to have a zero and near-zero AGB value in comparison to FIA plots. This issue can be related to issues of optical imagery, such as Landsat, which struggle to predict zero biomass values for green reflected areas without biomass. Moreover, it could be caused by FIA reference data labeled as "structural zeros", where they do not record biomass due to forest/nonforest definitions while trees exist there. All ecdf graphs at different scales for 2002 reach their maximum predicted values around 170 Mg/ha. For 2006 graph, at plot:pixel and 8660 ha scales, the ecdf reaches 150 Mg/ha, while at the 78,100 and 216,500 scales, it reaches 125 Mg/ha. For 2011, the maximum predicted value comes about 170 Mg/ha for both the plot:pixel and 8660 ha scale. Then, the maximum predicted value reaches 150 Mg/ha and 140 Mg/ha at 78,100 and 216,500 ha scales, respectively. At the plot:pixel and 8660 ha scales for 2016 and 2019, the maximum predicted value reaches 170 Mg/ha, whereas for the 78,100 and 216,500 ha scales, it is about150 Mg/ha. For the pooled datasets, the maximum predicted value is about 150 Mg/ha for the plot:pixel and 8660 ha scales and at 120 Mg/ha for the

78,100 and 216,500 ha scales. Reaching to the maximum predicted values for ecdf is a sign of underestimation in the RF model. Therefore, the AGB maps saturate sooner than the reference data. This could be a problem relating to both optical imagery and using airborne LiDAR rasters as training/testing sample. First, saturation is a common issue in optical imagery at regions with high biomass [17,53]. Second, in the process of generating airborne LiDAR rasters, underestimation of high biomass areas exist which seem to be inevitable.

Figure 11 represents the scatterplot comparison of modeled estimates versus estimates derived from the FIA plots at all four scales. The 1:1 line and the GMFR regression lines for each model–plot relationship are added to aid visual interpretation. The scatter about the GMFR line decreased with scale as expected, resulting in increasing $AC_u$ values with the scale as well (Tables 2–5). The $AC_s$ remain approximately the same at the plot:pixel and 8660 ha scales (Tables 2 and 3), while it decreases as the scale increases (Tables 4 and 5). It can be interpreted as the difference between the modeled AGB and the reference data. When the scale increases, samples with more variety are available and the systematic error decreases. Remaining steady and decreasing off the 1:1 line (Figure 11) indicates real error presented over and above-expected differences due to differences in spatial support and uncertainty in the reference datasets. This interpretation is more robust when the assessment includes scales with sufficient ground plots to be reasonably confident of reference data estimates. The distance and direction of the GMFR line from the 1:1 line provides an approximate indication of the direction and magnitude of the difference between the datasets at that scale. Moreover, if the GMFR line crosses the 1:1 line, it indicates that the direction of relative over- and underestimation changes at a threshold mean biomass/ha value [27]. These differences also may be due to the spatial pattern of biomass values in NY, in which high and low values are more spatially intermixed throughout the state.

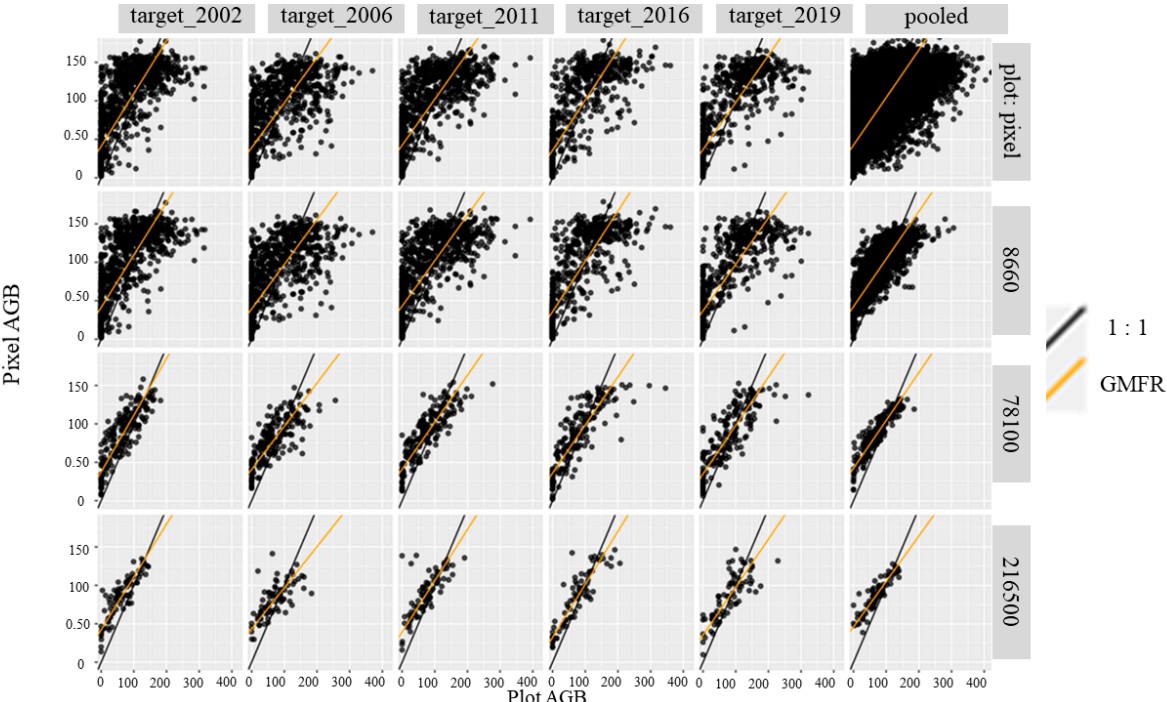

**Figure 11.** The 1:1 and GMFR lines across scales—FIA plots vs. mapped (SI corrected).

Results of Assessment 3: Figure 12 demonstrates the mapped differences in the modeled area estimates at the 216,500 ha scale with respect to plot-based confidence intervals for 2002, 2006, 2011, 2016, 2019, and the pooled datasets. In the 2002 map, about 60% of the time, the mean biomass estimates fall within the 95% confidence interval. About 66% of the time, the mean biomass estimates fall within the 95% confidence interval for 2006. As the 2011 distribution map indicates, approximately 72% of the time, the mean

biomass estimates fall within the 95% confidence interval. About 77% of the time, the mean biomass estimates fall within the 95% confidence interval for 2016. The 2019 mean biomass values fall within the 95% CI associated with the plot means of 86% of the time. For the pooled dataset, 28% of the time, the mean biomass estimates fall within the 95% confidence interval. In Figure 12, the largest errors occur outside the airborne LiDAR AGB pilot areas. It should be noted that training/testing datasets are within the four pilot areas; however, the proposed model predicts the AGB for the entire NYS.

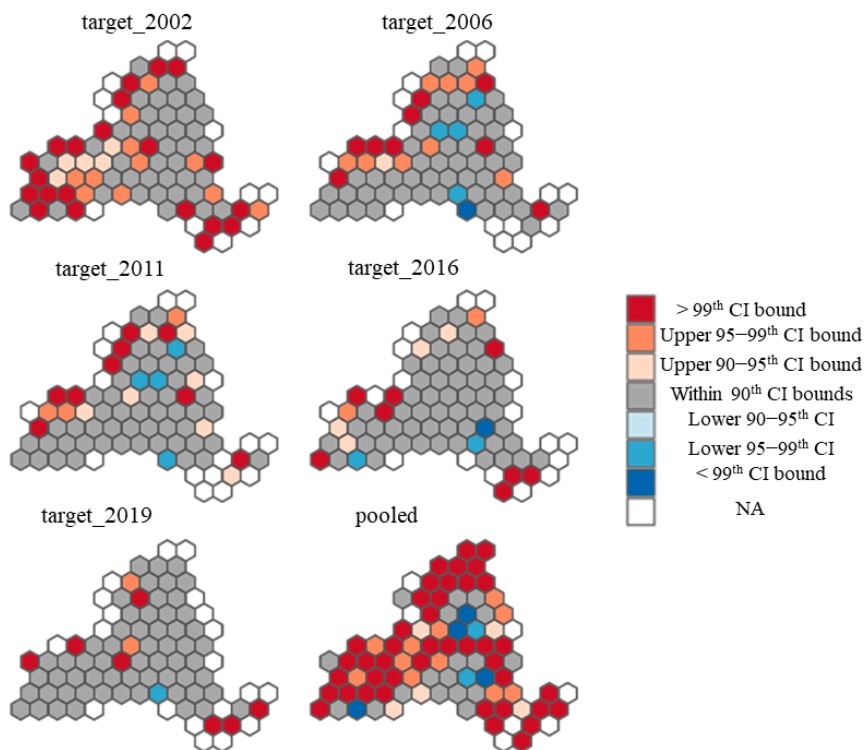

**Figure 12.** Mapped differences in modeled area estimates at the 216,500 ha scale with respect to plot-based confidence intervals (2002, 2006, 2011, 2016, 2019, and pooled).

Results of Assessment 4: Figure 13 shows the mapped results of local variability of plot AGB values in the 216,500 ha scale for 2002, 2006, 2011, 2016, 2019, and the pooled data in terms of the standard deviation. The local variability map of the FIA plot and the produced maps at the 216,500 ha scale for each year are shown separately. Generally, comparing the FIA and target maps shows that among all the maps, the local variability of the 2002 target map is more similar to the FIA plots than the others.

The assessment provided in this section presents useful information about the magnitude and direction of the error across the range of distribution of AGB values, such as Figure 11. Figure 12 shows the magnitude of the error in different regions of NYS. This assessment helps us to identify the type of the error. For instance, Figure 10 shows the frequency distribution of values and Figure 13 indicates the local spatial variability of the modeled geospatial dataset. Moreover, it can be analyzed what portion of the relative error could be attributed to systematic versus unsystematic differences in Tables 2–5.

Implementing assessments at different scales provides informative information which is related to the application. Finding the best scale is a matter of having sufficient reference data, which depends on the spatial intensity of the field measurements representing the application of the study. The ecdf graph assessment at different scale presents some information on which scale improves the model performance. However, at the same time, they need to be interpreted in association with the results from broader and finer scales. For example, when an error present at the 216,500 ha scale also occurs at finer scales, it is more likely to indicate that the difference observed at the finer scale contains a large component of real bias over and above uncertainty in the plot-based estimate.

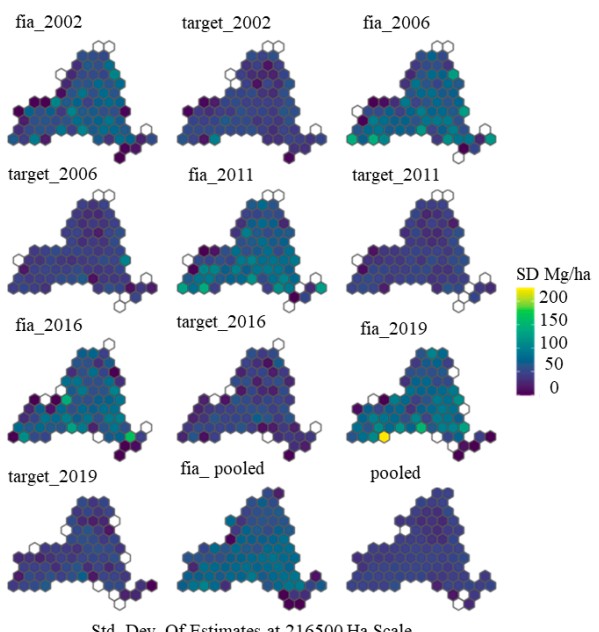

**Figure 13.** Mapped results of local variability of plot values in each 216,500 ha area (2002, 2006, 2011, 2016, 2019, and pooled).

## 5. Conclusions

This study proposed a framework for the historical AGB mapping of the entire NYS from 2001 to 2019 using the RF model in GEE. To achieve this goal, Landsat5 TM, Landsat 7 ETM, and Landsat 8 OLI imagery, along with environmental and topographical data, were utilized for AGB modeling using an object-based image analysis (OBIA) approach. For model training and validation, LiDAR-based AGB rasters from 2014, 2015, 2016, and 2018 were used. Then, the trained RF model was applied to generate the AGB maps for the remaining years. Results showed a decrease of $1289.70 \times 106$ Mg/ha in the AGB between 2001 and 2006, while the biomass increased about $838.2 \times 106$ Mg/ha from 2006 to 2011. An approximate AGB of 280.4 Mg/ha increased between 2011 and 2016, while the AGB difference between 2016 and 2019 was a decrease of 135.5 Mg/ha. In addition, map accuracy assessment of local variability, and spatial and distribution patterns of local differences provided information about the magnitude and direction of local bias. The results of the mapped differences in the modeled area estimates at the 216,500 ha scale showed that about 60%, 66%, 72%, 77%, 86%, and 28% of the time, the mean biomass estimates fall within the 95% confidence interval for 2002, 2006, 2011, 2016, 2019, and pooled datasets, respectively. The novelty of this study is associated with the use of the OBIA approach for accurate temporal state-wide AGB mapping. One of the limitations of this study is that, as these maps are generated based on the LiDAR raster, the uncertainty in the reference data can be prorogated throughout the workflow. However, the historical AGB maps can contribute to carbon stock monitoring over large areas, resulting in practical solutions for climate change issues.

**Author Contributions:** Conceptualization, B.S., M.M., C.M.B., H.T. and L.J.; supervision: B.S., C.M.B. and M.M.; formal analysis: H.T. and L.J.; data collection, H.T. and L.J.; visualization, H.T.; writing-original draft preparation, H.T.; writing-review and editing, B.S., M.M., C.M.B., H.T. and L.J. All the authors have read and agreed to the published version of the manuscript. All authors have read and agreed to the published version of the manuscript.

**Funding:** This project was financially supported by a McIntire-Stennis grant, funded through the USDA-NIFA (United States Department of Agriculture- National Institute of Food and Agriculture) and The Climate and Applied Forest Research Institute (CAFRI) at SUNY ESF funded through the NYS Department of Environmental Conservation.

**Data Availability Statement:** Not applicable.

**Acknowledgments:** This project was financially supported by a McIntire-Stennis grant, funded through the USDA-NIFA (United States Department of Agriculture- National Institute of Food and Agriculture) and The Climate and Applied Forest Research Institute (CAFRI) at SUNY ESF funded through the NYS Department of Environmental Conservation. We acknowledge both agencies for supporting this project.

**Conflicts of Interest:** The authors declare no conflict of interest.

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
