# Peer review of "Mapping Two Decades of New York State Forest Aboveground Biomass Change Using Remote Sensing"

_remotesensing, doi:10.3390/rs14164097_

Round 1

Reviewer 1 Report

This paper presents a framework to estimate forest aboveground biomass using Landsat imagery and GEE. The research is topical and the methodology is sound but there are a few areas that need improvement. Below are my comments and suggested edits.  

Title: I suggest removing “multi-source remote sensing data” from the title as the model is based on only Landsat data and LiDAR data were used for training and validation.

Introduction: It is good to better justify the advantages of using object-based approach for AGB estimation using medium resolution remote sensing data such as Landsat. What forest attributes can be better estimated that can contribute to AGB estimation.

Methods and Results:

Page 6 line 208: Please check TWI formula for correctness and also add equation number.

Page 6 line 210: DEM has already been spelled out, no need to do it again.

In Figure 2: LandTrendr was not included. Also it is not clear which variables were used to extract objects for OBIA.

Figure 1, Figure 4 and Figure 6: please correct Km to km in the scale bar.

Page 25 line 604: remove 6. Patents

Author Response

We thank you the editors and the reviewers for reviewing our manuscript and for providing useful comments. We believe the manuscript has been significantly improved after your comments. We revised the manuscript based on the reviewers’ comments. Changes are highlighted in red in the revised manuscript. In addition, below we provided a detailed response to each reviewer’s comments. Please see the attachment. 

Reviewer 2 Report

This paper presents an approach to combining Landsat and lidar data with forest inventory data, to estimate AGB at various scales, within NY state forest areas. The paper presents some potentially somewhat interesting results. But a lack of clarity in the method and analysis makes it v difficult to draw out the significance or utility of the approach at the moment. It's feasible that the authors could modify the work here sufficiently to make it acceptable for publication, but the paper does need quite some work for that to happen.   In particular, it's really not clear how they get AGB from the lidar data, despite mentioning the Jenkins et al allometry etc. This needs to be a lot clearer. Following that, there is a lack of clarity about how exactly the RF mapping has been carried out, what the sensitivity is to training data etc etc. There is also little consideration of uncertainty, so the final results are impossible to interpret unless they can put error bars on their AGB estimates. They mention uncertainty at times, but then don't do anything about it.   I have provided an annotated pdf that gives detailed comments on many of the aspects I raise generally here.

Author Response

We thank you the editors and the reviewers for reviewing our manuscript and for providing useful comments. We believe the manuscript has been significantly improved after applying your comments. We revised the manuscript based on the reviewers’ comments. Changes are highlighted in red in the revised manuscript. In addition, below we provided a detailed response to each reviewer’s comments.

Author Response

(The authors gave the same response as above.)

Round 2

Reviewer 2 Report

I think the authors have done a reasonable job of addressing some of my comments from the previous review. However I still think there are significant issues of method that are either not clear or not justified. 

In particular the authors state in a number of places that "Landsat imagery can be used directly to estimate the above-ground biomass." eg comment 8. This is just not true. Landsat can be used to very indirectly estimate very low values of AGB, and even this assumes that the spectral response is sensitive to biomass, which above a very low AGB threshold it just isn't. This is why eg canopy or tree height is needed, or SAR backscatter. In the Li et al 2021 paper they cite in support of their argument, the regression is based on Landsat AND Sentinel-1 SAR data. The authors really have to rethink their argument here. 

Response 15 and 24 - it still isn't clear what they mean by data collection between July and August. See figure in comment 24: "We meant image composite between 07/01/2011 to 08/31/2011. We changed the caption of the figure to be more clear."  Composite of what though, how? How many images, composited how - mean values, what ... ? This is still not clear at all.

Comment 19 - so it's still not clear but are the lidar data ONLY being used to calculate a DEM? I may have misunderstood but if so, then please make this clear. But that also means that the single source of RS data here that is capable of being used to estimate AGB most directly, is not actually being used for that. Which seems v odd. 

Comment 21: "The authors meant that after some iterations most of the parameters remain same, thus; they are chosen as final values of parameter tuning" - what is some iterations, how many, why that many? What does 'most of' mean? This is all still vague.